# REGULARISED JUMP MODELS FOR REGIME IDENTIFICATION AND FEATURE SELECTION

## ABSTRACT

A regime modelling framework can be employed to address the complexities of financial markets. Under the framework, market periods are grouped into distinct regimes, each distinguished by similar statistical characteristics. Regimes in financial markets are not directly observable but are often manifested in market and macroeconomic variables. The objective of regime modelling is to accurately identify the active regime from these variables at a point in time, a process known as *regime identification*. One way to enhance the accuracy of regime identification is to select features that are most responsible for statistical differences between regimes, a process known as *feature selection*. Models based on the Jump Model framework have recently been developed to address the joint problem of regime identification and feature selection. In the following work, we propose a new set of models called *Regularised Jump Models* that are founded upon the Jump Model framework. These models perform feature selection that is more interpretable than that from the Sparse Jump Model, a model proposed in the literature pertaining to the Jump Model framework. Through a simulation experiment, we find evidence that these new models outperform the Standard and Sparse Jump Models, both in terms of regime identification and feature selection.

## 1 INTRODUCTION

The section outlines and examines the constituent models of the Jump Model framework introduced in (2). A new set of models called *Regularised Jump Models* are then introduced and compared to the existing models.

### 1.1 STANDARD JUMP MODEL

Denote the state or mode sequence associated with the sequence of observations $\boldsymbol{Y}$ by $(s_1, s_2, \ldots, s_T)$ where each $s_t \in \{1, 2, \ldots, K\}$ for $t = 1, 2, \ldots, T$. $K$ is the number of states and is assumed to be known. The $K$ model parameters are given by $\boldsymbol{\mu}_1, \boldsymbol{\mu}_2, \ldots, \boldsymbol{\mu}_K$ where each $\boldsymbol{\mu}_k \in \mathbb{R}^p$ for $k = 1, 2, \ldots, K$. The model parameters are the conditional means of the features assigned to each of the $K$ states, hence the notational choice $\boldsymbol{\mu}_k, k = 1, 2, \ldots, K$.

**Definition 1.1** (Standard Jump Model). The *Standard Jump Model* with $K$ states is defined by the minimisation of the objective function

$$\sum_{t=1}^{T-1} \left[ \|\boldsymbol{y}_t - \boldsymbol{\mu}_{s_t}\|^2 + \lambda \mathbb{1}_{\{s_t \neq s_{t+1}\}} \right] + \|\boldsymbol{y}_T - \boldsymbol{\mu}_{s_T}\|^2 \tag{1.1}$$

over the model parameters $\boldsymbol{\mu}_1, \boldsymbol{\mu}_2, \ldots, \boldsymbol{\mu}_K$ and state sequence $(s_1, s_2, \ldots, s_T)$. The term $\|\boldsymbol{y}_t - \boldsymbol{\mu}_{s_t}\|^2$ represents the squared $\mathcal{L}_2$-distance between the vectors $\boldsymbol{y}_t$ and $\boldsymbol{\mu}_{s_t}$ and $\lambda \geq 0$ is a hyperparameter.

The objective function in (1.1) can be interpreted as a tradeoff between model fitting and prior assumptions about the tendency of the sequence $S$ to "jump", or change states. This tendency is

controlled by the hyperparameter $\lambda$. When $\lambda = 0$, the model reduces to splitting the dataset in at most $K$ states and fitting one model per state, thereby generalising the K-means algorithm ((9)). For $\lambda \to \infty$, the Standard Jump Model results in a single-state model since the cost of changing states becomes prohibitive.

## 1.2 Sparse Jump Model

The Sparse Jump Model is an extension of the Standard Jump Model in its incorporation of feature selection. Let $\boldsymbol{w} := (w_1, w_2, \ldots, w_p) \in \mathbb{R}^p$ denote a vector of feature weights that are assumed to be the same across all states.

Feature selection is incorporated in the Sparse Jump Model by employing the criterion in (13). The criterion is given by

$$
\begin{aligned}
\max \quad & \boldsymbol{w}' \left( \sum_{k=1}^{K} |C_k| \left(\boldsymbol{\mu}_k - \bar{\boldsymbol{\mu}}\right)^2 \right), \\
\text{such that} \quad & \|\boldsymbol{w}\|^2 \leq 1, \quad \|\boldsymbol{w}\|_1 \leq \kappa, \\
& w_p \geq 0 \; \forall p,
\end{aligned}
\tag{1.2}
$$

with respect to the parameters $\boldsymbol{\mu}_1, \boldsymbol{\mu}_2, \ldots, \boldsymbol{\mu}_K$, state sequence $(s_1, s_2, \ldots, s_T)$ and feature weights $w_1, w_2, \ldots, w_p$. The term $|C_k| \left(\boldsymbol{\mu}_k - \bar{\boldsymbol{\mu}}\right)^2$ in (1.2) is a vector of size $p$ whose entries are the contributions of each feature to the between-cluster sum of squares (BCSS) in the $k^{\text{th}}$ cluster.

If we consider the clusters $C_1, C_2, \ldots, C_K$ fixed, then the feature weights will be assigned to features based on their individual BCSS contributions: features with larger BCSS contributions will be given larger weights which, in turn, optimises the overall spread between the clusters.

$\kappa \in [1, \sqrt{p}]$ in (1.2) is a hyperparameter that controls the degree of sparsity in the feature weights. The squared $\mathcal{L}_2$ penalty in (1.2) serves an important role since without it, at most one element of $\boldsymbol{w}$ would be non-zero in general when features are correlated (see (14) for more details). If $w_1 = w_2 = \cdots = w_p$, then 1.2 reduces to the maximisation of the BCSS (equivalent to the minimisation of the within-cluster sum of squares (WCSS)) which is the objective of the K-means clustering algorithm.

Combining 1.2 with a jump penalty, we give the below definition for the Sparse Jump Model:

**Definition 1.2** (Sparse Jump Model). The *Sparse Jump Model* with $K$ states is defined by the optimisation programme

$$
\begin{aligned}
\max \quad & \boldsymbol{w}' \left( \sum_{k=1}^{K} |C_k| \left(\boldsymbol{\mu}_k - \bar{\boldsymbol{\mu}}\right)^2 \right) - \lambda \sum_{t=1}^{T} \mathbb{1}_{\{s_t \neq s_{t+1}\}} \\
\text{such that} \quad & \|\boldsymbol{w}\|^2 \leq 1, \quad \|\boldsymbol{w}\|_1 \leq \kappa, \\
& w_p \geq 0 \; \forall p,
\end{aligned}
\tag{1.3}
$$

,

with respect to the parameters $\boldsymbol{\mu}_1, \boldsymbol{\mu}_2, \ldots, \boldsymbol{\mu}_K$, state sequence $(s_1, s_2, \ldots, s_T)$ and feature weights $w_1, w_2, \ldots, w_p$. $\kappa \in [1, \sqrt{p}]$ in (1.2) is a hyperparameter that controls the degree of sparsity in the feature weights.

If $w_1 = w_2 = \ldots, w_p$, then Definition 1.2 reduces to the Standard Jump Model in Definition 1.1.

## 1.3 Regularised Jump Models

In this section, a new approach to feature selection in the Jump Model framework is introduced. The approach is an adaptation of the Regularised K-means algorithm proposed in (11) to the Jump Model framework. Therefore, we call the models constituting this approach *Regularised Jump Models*.

## 1.4 Regularised K-Means

**Definition 1.3.** The *Regularised K-means* algorithm is defined by the minimisation of

$$\sum_{k=1}^{K} \left\{ \sum_{t \in C_k} \| \boldsymbol{y}_t - \boldsymbol{\mu}_k \|^2 \right\} + \gamma \mathcal{P} \left( \underline{\boldsymbol{\mu}} \right), \tag{1.4}$$

with respect to the clusters $C_1, C_2, \ldots, C_K$ and matrix of cluster centres $\underline{\boldsymbol{\mu}} \in \mathbb{R}^{K \times p}$. $\gamma \geq 0$ is a tuning parameter that controls the amount of regularisation applied to the cluster centres. $\mathcal{P} : \mathbb{R}^{K \times p} \to \mathbb{R}$ is a penalty function that depends on $\underline{\boldsymbol{\mu}}$. The first term is the objective function of standard K-means clustering introduced in (8).

From (11), below are several penalty function options which are named after their counterparts from regularised regression:

$$\mathcal{L}_0 : \quad \mathcal{P}_0 \left( \underline{\boldsymbol{\mu}} \right) = \sum_{j=1}^{p} \mathbf{1}_{\{\| \boldsymbol{\mu}_{.,j} \| > 0\}} \tag{1.5a}$$

$$\text{Lasso} : \quad \mathcal{P}_1 \left( \underline{\boldsymbol{\mu}} \right) = \sum_{j=1}^{p} \| \boldsymbol{\mu}_{.,j} \|_1 \tag{1.5b}$$

$$\text{Ridge} : \quad \mathcal{P}_2 \left( \underline{\boldsymbol{\mu}} \right) = \sum_{j=1}^{p} \| \boldsymbol{\mu}_{.,j} \|^2 \tag{1.5c}$$

$$\text{Group-Lasso} : \quad \mathcal{P}_3 \left( \underline{\boldsymbol{\mu}} \right) = \sum_{j=1}^{p} \| \boldsymbol{\mu}_{.,j} \|, \tag{1.5d}$$

where $\boldsymbol{\mu}_{.,j}$ is the $j^{\text{th}}$ column of $\underline{\boldsymbol{\mu}}$. The penalty on $\underline{\boldsymbol{\mu}}$ balances the size of the cluster centres and their contribution to the objective function in (1.4).

The intuition for penalising the size of the cluster centres lies in the fact that when a variable does not contribute to the partitioning of the data, its estimated cluster centres will be close to the overall mean of the data (see Proposition 1 in (11) for more details).

## 1.5 Regularised Jump Model Equation

Combining (1.4) with a jump penalty, we propose the following definition for the Regularised Jump Model.

**Definition 1.4** (Regularised Jump Model)**.** The *Regularised Jump Model* with $K$ states is defined by the minimisation of the objective function

$$\sum_{t=1}^{T-1} \left\{ \| \boldsymbol{y}_t - \boldsymbol{\mu}_{s_t} \|^2 + \lambda \mathbf{1}_{\{s_t \neq s_{t+1}\}} \right\} + \| \boldsymbol{y}_T - \boldsymbol{\mu}_{s_T} \|^2 + \gamma \mathcal{P} \left( \underline{\boldsymbol{\mu}} \right), \tag{1.6}$$

with respect to $\underline{\boldsymbol{\mu}}$ and the state sequence $(s_1, s_2, \ldots, s_T)$. $\lambda, \gamma \geq 0$ are hyperparameters and the penalty function $\mathcal{P} : \mathbb{R}^{K \times p} \to \mathbb{R}$ can be chosen from those listed in (1.5).

$\gamma = 0$ reduces Definition 1.4 to the Standard Jump Model in Definition 1.1. $\lambda = 0$ reduces Definition 1.4 to the Regularised K-Means model in Definition 1.3.

## 2 Calibration of Jump Models

The section details the algorithms that perform calibration of the Jump Models outlined in the previous section. The models are then tested in a simulation experiment.

## 2.1 Standard Jump Model Calibration

Denote $\underline{\boldsymbol{\mu}} := (\boldsymbol{\mu}_1, \boldsymbol{\mu}_2, \ldots, \boldsymbol{\mu}_K)' \in \mathbb{R}^{K \times p}$ the matrix of model parametersand the state sequence $S := (s_1, s_2, \ldots, s_T)$. The calibration algorithm for the Standard Jump Model was proposed in (10)

and is shown in Algorithm 1. The model is calibrated using a coordinate descent algorithm that alternates between finding the model parameters $\boldsymbol{\mu}_1, \boldsymbol{\mu}_2, \ldots, \boldsymbol{\mu}_K$ that minimise the objective function (1.1) with a fixed state sequence and finding the state sequence $(s_1, s_2, \ldots, s_T)$ that minimise the objective function (1.1) with fixed $\boldsymbol{\mu}_1, \boldsymbol{\mu}_2, \ldots, \boldsymbol{\mu}_K$.

Following the algorithm proposed in (10), the process is repeated ten times at the most or until the state sequence does not change after one iteration. However, there is no guarantee that the solution reached is the global solution since the solution depends on the initial state sequence.

Therefore, we adopt the initialisation method in (10): the coordinate descent algorithm is run from ten different state sequences in parallel and the model that achieves the lowest objective function value is chosen. These initial state sequences are generated by the K-means++ seeding technique introduced in (1) which has been shown to improve both the speed and accuracy of standard K-means clustering.

We note that the objective function in Step 5.1 of Algorithm 1 is convex in $\underline{\boldsymbol{\mu}}$ and can be solved in closed-form. $\boldsymbol{\mu}_k \in \mathbb{R}^p$ has the below optimal solution at the $j^{\text{th}}$ iteration:

$$\boldsymbol{\mu}_k^{(j)} = \frac{\sum_{t=1}^{T} \boldsymbol{y}_t \mathbf{1}_{\{s_t^{(j-1)}=k\}}}{\sum_{t=1}^{T} \mathbf{1}_{\{s_t^{(j-1)}=k\}}}, \ k = 1, 2, \ldots, K.$$

$S^{(j)}$ in Step 5.2 is obtained using the following dynamic programming equations. Define

$$V(T, s) = \|\boldsymbol{y}_T - \boldsymbol{\mu}_s\|^2, \tag{2.1a}$$
$$V(t, s) = \|\boldsymbol{y}_t - \boldsymbol{\mu}_s\|^2 + \min_j \left\{ V(t+1, j) + \lambda \mathbf{1}_{\{s \neq j\}} \right\}, \tag{2.1b}$$

for $t = T - 1, \ldots, 2, 1$. The most likely state sequence is then given by

$$s_1 = \arg\min_j V(1, j), \tag{2.2a}$$
$$s_t = \arg\min_j \left\{ V(t, j) + \lambda \mathbf{1}_{\{s_{t-1} \neq j\}} \right\}, \ t = 2, \ldots, T. \tag{2.2b}$$

## 2.2 Sparse Jump Model Calibration

The calibration of the Sparse Jump Model can be performed using an extension of the coordinate descent algorithm in Algorithm 1. Holding the feature weight vector $\boldsymbol{w}$ fixed, (1.3) is optimised in terms of $\underline{\boldsymbol{\mu}}$ and $S$. Secondly, holding $\underline{\boldsymbol{\mu}}$ and $S$ fixed, (1.3) is optimised in terms of $\boldsymbol{w}$.

As noted in (9), this iterative approach is not guaranteed to generate a global optimum because the problem is non-convex. Additionally, the first optimisation involves applying the fitting algorithm of the Standard Jump Model to a weighted version of the data, which by itself is not guaranteed to find a global optimum.

In Step 2(d), solving (1.3) with respect to $\boldsymbol{w}$, while keeping $\underline{\boldsymbol{\mu}}$ and $S$ fixed, can be done using soft-thresholding. The soft-threholding operator $S$ is given by $S(\boldsymbol{x}, c) = \text{sgn}(\boldsymbol{x}) \odot (|\boldsymbol{x}| - c)_+$, where $\boldsymbol{x}_+$ denotes the positive part of the elements in $\boldsymbol{x}$ and $\odot$ denotes element-wise multiplication.

The solution to the convex problem (1.3), which follows from the Karush-Kuhn-Tucker conditions (more details can be found in (4)), is $\boldsymbol{w} = \frac{S(\boldsymbol{x}, \Delta)}{\|S(\boldsymbol{x}, \Delta)\|}$ where $\boldsymbol{x} = \sum_{k=1}^{K} |C_k| (\boldsymbol{\mu}_k - \bar{\boldsymbol{\mu}})^2$ is a vector comprising the between-cluster sum of squares (BCSS) contributions of each feature.

Here, $\Delta = 0$ if that results in $\|\boldsymbol{w}\|_1 \leq \kappa$; otherwise, we choose $\Delta > 0$ to yield $\|\boldsymbol{w}\|_1 = \kappa$. This assumes that there is a unique maximal element of $\boldsymbol{x}$ and that $1 \leq \kappa \leq \sqrt{p}$ (13).

## 2.3 Regularised Jump Model Calibration

We propose calibrating the Regularised Jump Models using a coordinate descent algorithm similar to Algorithm 1 for Standard Jump Models. The coordinate descent algorithm alternates between

---

**Algorithm 1** Calibration of Standard Jump Model in (10)

---

**Input:** Training dataset $\boldsymbol{Y} = (\boldsymbol{y}_1, \boldsymbol{y}_2, \ldots, \boldsymbol{y}_T)$, assumed number of states $K$, and jump penalty $\lambda$.

---

**Step 1**: Initialise state sequence $S^{(0)} := \left( s_1^{(0)}, s_2^{(0)}, \ldots, s_T^{(0)} \right)$.

**Step 2**: Iterate for $j = 1, 2, \ldots, 10$:

   Fit model parameters $\boldsymbol{\mu}^{(j)}$:

$$\boldsymbol{\mu}^{(j)} \leftarrow \arg \min_{\boldsymbol{\mu}} \sum_{t=1}^{T} \| \boldsymbol{y}_t - \boldsymbol{\mu}_{s_t^{(j-1)}} \|^2. \tag{5.1}$$

   Fit state sequence $S^{(j)}$:

$$S^{(j)} \leftarrow \arg \min_{S} \left\{ \sum_{t=1}^{T-1} \left\{ \| \boldsymbol{y}_t - \boldsymbol{\mu}_{s_t}^{(j)} \|^2 + \lambda \mathbb{1}_{\{s_t \neq s_{t+1}\}} \right\} + \| \boldsymbol{y}_T - \boldsymbol{\mu}_{s_T}^{(j)} \|^2 \right\}. \tag{5.2}$$

   Break if $S^{(j-1)} = S^{(j)}$.

---

**Output**: Estimated model parameters $\boldsymbol{\mu}^*$ and state sequence $S^*$.

---

minimising (1.6) with respect to the state sequence $\boldsymbol{\mu}$ while keeping $S$ fixed, and minimising (1.6) with respect to $S$ while keeping $\boldsymbol{\mu}$ fixed. These two steps are repeated for ten iterations at the most or until $S$ does not change after two consecutive iterations.

Firstly, seven initial state sequences are generated using the strategy proposed in (11) which the authors show to be the most optimal for the Regularised K-Means model after benchmarking it against popular initialisation strategies. Given the feature selection aspect of the Regularised Jump Model, the initialisation strategy incorporates potential sparsity in the initial cluster centres. The initialisation strategy is given in Algorithm 3.

Each initial state sequence outputted from Algorithm 3 is used as an input in Algorithm 4. The algorithm is run in parallel using these seven initial cluster assignments and the run which ultimately yield the lowest objective function value, is chosen as the final result.

Once an initial state sequence has been generated, the model parameters $\boldsymbol{\mu}$ are updated (Step 8.1). In particular, assuming a fixed state sequence $S = (s_1, s_2, \ldots, s_T)$, the following problem is solved:

$$\arg \min_{\boldsymbol{\mu}} \sum_{t=1}^{T} \| \boldsymbol{y}_t - \boldsymbol{\mu}_{s_t} \|^2 + \gamma \mathcal{P}\left( \boldsymbol{\mu} \right). \tag{2.3}$$

The solution to (2.3) for each penalty function introduced in (1.5), is given in B.1. Once $\boldsymbol{\mu}$ is updated, the state sequence $S$ is updated.

Since Steps 8.2 and 5.2 are identical, the state sequence is updated using the same dynamic programming equations in (2.1) and (2.2).

## 3 HYPERPARAMETER TUNING

Table 1 shows the hyperparameters of each Jump Model.

The literature on the hyperparameter tuning of Jump Models is limited. (5) suggests tuning based on a grid search of potential hyperparmeter values and picking which combination produces the lowest Fang-Tang Information Criterion (FTIC) value, a criterion introduced in (6).

---

**Algorithm 2** Calibration of Sparse Jump Model in (9)

---

**Input:** Training dataset $\boldsymbol{Y} = (\boldsymbol{y}_1, \boldsymbol{y}_2, \ldots, \boldsymbol{y}_T)$, assumed number of states $K$ and hyperparameters $\lambda$ and $\kappa$.

---

**Step 1**: Initialise feature weights $\boldsymbol{w}$ as $\boldsymbol{w}^{(0)} = \left( \frac{1}{\sqrt{p}}, \frac{1}{\sqrt{p}}, \ldots, \frac{1}{\sqrt{p}} \right)$.

**Step 2**: Iterate for $i = 1, 2, \ldots$, until $\|\boldsymbol{w}^{(i)} - \boldsymbol{w}^{(i-1)}\|_1 / \|\boldsymbol{w}^{(i-1)}\|_1 < 10^{-4}$:

(a) Compute sequence of weighted features

$$\boldsymbol{z}_t = \boldsymbol{y}_t \odot \sqrt{\boldsymbol{w}^{(i-1)}}, \ t = 1, 2, \ldots, T,$$

where $\sqrt{\boldsymbol{w}^{(i)}}$ is the element-wise square root of $\boldsymbol{w}^{(i)}$.

(b) Initialise state sequence $S^{(0)} := \left( s_1^{(0)}, s_2^{(0)}, \ldots, s_T^{(0)} \right)$.

(c) Iterate for $j = 1, 2, \ldots, 10$:

i. Fit model parameters:

$$\underline{\boldsymbol{\mu}}^{(j)} \leftarrow \arg\min_{\underline{\boldsymbol{\mu}}} \sum_{t=1}^{T} \| \boldsymbol{z}_t - \boldsymbol{\mu}_{s_t^{(j-1)}} \|^2.$$

ii. Fit state sequence:

$$S^{(j)} \leftarrow \arg\min_{S} \left\{ \sum_{t=1}^{T} \left\{ \| \boldsymbol{z}_t - \boldsymbol{\mu}_{s_t}^{(j)} \|^2 + \lambda \mathbf{1}_{\{s_t \neq s_{t+1}\}} \right\} + \| \boldsymbol{z}_T - \boldsymbol{\mu}_{s_T}^{(j)} \|^2 \right\}.$$

Break if $S^{(j-1)} = S^{(j)}$.

(d) Update weights $\boldsymbol{w}^{(i)}$, while holding the model parameters $\underline{\boldsymbol{\mu}}^{(j)}$ and $S^{(j)}$ fixed, by solving (1.3) using soft thresholding.

---

**Output**: Estimated model parameters $\underline{\boldsymbol{\mu}}^*$, state sequence $S^*$ and feature weights $\boldsymbol{w}^*$.

---

---

**Algorithm 3** Initialisation of state sequences for calibration of Regularised Jump Model in (11)

---

**Step 1** Cluster $\boldsymbol{Y} = (\boldsymbol{y}_1, \boldsymbol{y}_2, \ldots, \boldsymbol{y}_T)$ using standard K-means, obtaining a matrix of initial cluster centres $\underline{\boldsymbol{\mu}} = (\boldsymbol{\mu}_1, \boldsymbol{\mu}_2, \ldots, \boldsymbol{\mu}_K)'$.

**Step 2** Compute the Euclidean norm for each cluster centre $d_j = \|\boldsymbol{\mu}_{.,j}\|$ for $j = 1, 2, \ldots, p$ and order them in descending order.

**Step 3** Execute K-means on the subset of variables corresponding to the 1, 2, 5, 10, 25, 50 and 100% largest $d_j$.

---

**Output**: Seven initial state sequences $\left( s_1^1, s_2^1, \ldots, s_T^1 \right), \left( s_1^2, s_2^2, \ldots, s_T^2 \right), \ldots, \left( s_1^7, s_2^7, \ldots, s_T^7 \right)$.

---

---

**Algorithm 4** Coordinate descent algorithm for calibration of Regularised Jump Model

---

**Input:** Training dataset $\boldsymbol{Y} = (\boldsymbol{y}_1, \boldsymbol{y}_2, \ldots, \boldsymbol{y}_T)$, assumed number of states $K$, penalty function $\mathcal{P}$, hyperparameters $\lambda$ and $\gamma$, and initial state sequence $S^{(0)} := \left( s_1^{(0)}, s_2^{(0)}, \ldots, s_T^{(0)} \right)$ generated using Algorithm 3.

---

**Step 1**: Iterate for $k = 1, 2, \ldots, 10$:

$$\underline{\boldsymbol{\mu}}^{(k)} \leftarrow \arg\min_{\underline{\boldsymbol{\mu}}} \sum_{t=1}^{T} ||\boldsymbol{y}_t - \boldsymbol{\mu}_{s_t^{(k-1)}}||^2 + \gamma \mathcal{P}\left( \underline{\boldsymbol{\mu}} \right), \tag{8.1}$$

$$S^{(k)} \leftarrow \arg\min_{S} \left\{ \sum_{t=1}^{T-1} \left\{ \|\boldsymbol{y}_t - \boldsymbol{\mu}_{s_t}^{(k)}\|^2 + \lambda \mathbb{1}_{\{s_t \neq s_{t+1}\}} \right\} + \|\boldsymbol{y}_T - \boldsymbol{\mu}_{s_T}^{(k)}\|^2 \right\}. \tag{8.2}$$

Break if $S^{(k)} = S^{(k-1)}$.

---

**Output**: Estimated matrix of cluster centres (or model parameters) $\underline{\boldsymbol{\mu}}^*$ and state sequence $S^*$.

---

| **Model** | $\lambda$ | $\kappa$ | $\gamma$ |
|---|---|---|---|
| Standard Jump | ✓ | | |
| Sparse Jump | ✓ | ✓ | |
| Regularised Jump | ✓ | | ✓ |

Table 1: Jump Model hyperparameters

In (5), the FTIC equation is an approximation and applies only to the Standard and Sparse Jump Models. Approximating information criteria such as the FTIC would require knowledge of the likelihood function of the various Jump Models which, for the Regularised Jump Models, is unavailable.

Instead, we opt for a non-parametric criterion that is applicable to all Jump Models. We propose hyperparameter tuning based on a criterion pertaining to *clustering stability*. The key idea of clustering stability is that if we repeatedly draw samples from the same distribution as that of the original dataset, and apply the Jump Model calibration algorithms, a good algorithm should produce state sequences that are similar from one sample to another.

Following the notation in (12) to some extent, denote $Z = \{X_1, X_2, \ldots, X_T\}$ a random sample of size $T$ from some unknown distribution function $F : \mathbb{R}^p \to \mathbb{R}$. Let $\boldsymbol{\theta}$ denotes the vector of active hyperparameters for a given Jump Model which can be referenced in Table 1. A clustering assignment $\psi(x)$ is defined as a mapping $\psi : \mathbb{R}^p \to \{1, 2, \ldots, K\}$ and a given Jump Model generates a clustering assignment $\Psi(., \boldsymbol{\theta})$ when applied to a sample $Z$.

**Definition 3.1** (Clustering Distance)**.** The *clustering distance* between any two clustering assignments $\psi_1(x)$ and $\psi_2(x)$ is defined as

$$d(\psi_1, \psi_2) = \mathbb{P}\left( \{\psi_1(X) = \psi_1(Y)\} \cap \{\psi_2(X) = \psi_2(Y)\} \right),$$

where $X, Y \in \mathbb{R}^p$ are realisations sampled from $F$.

The distance between $\psi_1$ and $\psi_2$ measures the probability of their disagreement. The clustering instability of a given Jump Model is given in the following definition:

**Definition 3.2** (Clustering Instability)**.** The *clustering instability* of a clustering algorithm $\Psi$ is given by

$$S(\Psi; \boldsymbol{\theta}, T) = \mathbb{E}[d(\Psi(Z_1; \boldsymbol{\theta}), \Psi(Z_2; \boldsymbol{\theta}))], \tag{3.1}$$

where $\Psi(Z_1; \boldsymbol{\theta})$ and $\Psi(Z_2; \boldsymbol{\theta})$ are two clustering assignments obtained by applying $\Psi(.; \boldsymbol{\theta})$ to $Z_1$ and $Z_2$ which are two independent samples from $F$ of size $T$.

Hyperparameter tuning is thus performed by finding a set of hyperparameter values that minimise (3.1). Various methods of estimating (3.1) for hyperparameter tuning in clustering problems have been proposed ((12), (7) and (3)).

We opt for a simple method of estimating (3.1) that is similar to the one proposed in (7); ten bootstrapped samples of the dataset $Y$ are generated, each of which are size $T$. For a fixed combination of hyperparameter values and a given Jump Model, ten clustering assignments (or state sequences) are estimated once the model is fitted onto each bootstrapped sample. The number of times any two state sequences are not equal, are then counted and averaged across all $\binom{10}{2} = 45$ comparisons. This estimate can be expressed mathematically as

$$\hat{S}\left(\Psi; \boldsymbol{\theta}, T\right) = \binom{10}{2}^{-1} \sum_{i=1}^{10} \sum_{j=i+1}^{10} \left(\sum_{t=1}^{T} \mathbf{1}_{\{\hat{s}_t^i \neq \hat{s}_t^j\}}\right), \tag{3.2}$$

where $\hat{s}_t^i$ is the estimated state at time $t$ for the $i^{\text{th}}$ boostrapped sample.

## 4 SIMULATION STUDY

We conduct a simulation study to compare the accuracy of the Standard, Sparse and Regularised Jump Models, with respect to both state estimation and feature selection. In the simulation study, the true state sequence and array of relevant features are both known, which makes it possible to evaluate the ability of each model to correctly identify the state sequence and set of relevant features.

We adopt the same data generating process as in (9). In their simulation study, the authors test the Standard and Sparse Jump Models against several popular regime-switching models such as the Hidden Markov Model (HMM) and its various extensions, and determine that these two Jump Models deliver superior performance.

Instead of duplicating the simulation study by testing the same regime-switching models, we test the proposed Regularised Jump Models and compare their performances with the Standard and Sparse Jump Models.

### 4.1 RESULTS

Tables 2 and 3 compare the respective state estimation and feature selection performances of the Standard Jump Model (Standard), Sparse Jump Model (Sparse) and Regularised Jump Model with $\mathcal{P}_k$ Penalty Function (Regularised $\mathcal{P}_k$) for $k \in \{0, 1, 2, 3\}$, for different values of $\mu$ and for different numbers of features $P$. The reported values of Tables 2 and 3 are the mean (and standard deviation) of the BAC of the estimated state sequence and of (3.2) respectively; the means and standard deviations are calculated over 100 simulations of $T = 500$ observations for each combination of $\mu$ and $P$.

Similar to (9), we use the Wilcoxon signed-rank test to determine whether the differences between the BACs in the Regularised Jump Models and those in the Sparse Jump Model are statistically significant. Bold entries in Tables 2 and 3 denote BACs of a Regularised Jump Model that are higher than that of the Sparse Jump Model with statistical significance $\alpha = 0.05$.

Comparing Standard and the remaining models in Table 2, it is evident that feature selection improves accuracy of state estimation when the number of features $P$ is increased. In Table 2, the BAC of the Standard Jump Model, a model that does not perform feature selection, generally decreases when $P$ is increased. On the other hand, the BAC of the Sparse and Regularised Jump Models stay approximately level compared to the case of there being no irrelevant features (when $P = 15$).

The feature selection is more efficient for higher values of $\mu$ since for higher values of $\mu$, the relevance of the first fifteen features becomes more apparent and hence easier for the model to identify. This logic is reinforced by the results in Table 3; the feature selection performance of the models improves for higher values of $\mu$.

When comparing the relative performances of the Sparse, Regularised $\mathcal{P}_0$, $\mathcal{P}_1$ and $\mathcal{P}_3$ Jump Models in Tables 2 and 3, we can see a correspondence between classification accuracy in terms of state

estimation, and classification accuracy in terms of feature selection: models that are more accurate in terms of feature selection, are also more accurate in terms of state estimation. This further highlights the importance of feature selection in accurately estimating the true state sequence.

If we unbundle the BACs in Tables 2 and 3 in terms of $\mu$, model and number of features $P$, and calculate the Spearman rank correlation coefficient between these two unbundled series of BAC values, we compute an estimate of 91% with a $p$-value of $\mathcal{O}\left(10^{-24}\right)$. This leads us to reject the null hypothesis of there being no monotonic relationship between state estimation accuracy and feature selection accuracy.

Tables 2 and 3 demonstrate outperformance of the Regularised $\mathcal{P}_1, \mathcal{P}_2$ and $\mathcal{P}_3$ Models compared to both the Standard and Sparse Jump Models. The Regularised $\mathcal{P}_0$ model performs roughly in line with, or slight worse than, the Sparse Jump Model in terms of state estimation and feature selection.

In Table 2, the outperformance of the Regularised $\mathcal{P}_1, \mathcal{P}_2$ and $\mathcal{P}_3$ is most evident for the cases $\mu \geq 0.5$ and $P < 300$. For $\mu = 0.25$ and $P = 300$, all models produce roughly the same BACs that range between 0.334 and 0.344.

Similar results can be observed in Table 3. All models perform at approximately the same level for $\mu = 0.25$ and for $\mu \geq 0.5$, the Regularised $\mathcal{P}_1$ and $\mathcal{P}_3$ outperform both the Sparse and Regularised $\mathcal{P}_0$ models. The outperformance of the Regularised $\mathcal{P}_1, \mathcal{P}_2$ and $\mathcal{P}_3$ over the Sparse Jump Model is statistically significant for some cases, as shown by the bold entries in Tables 2 and 3.

## 5 CONCLUSION

In the new class of models called *Regularised Jump Models*, the regularised K-means model proposed in (11) has been adapted to the Jump Model Framework. These models perform joint state and parameter estimation, and feature selection. The feature selection process performed by the Regularised Jump Models is more direct and interpretable than that from the Sparse Jump Model, which performs feature selection by proxy. These models were tested in a simulation experiment and demonstrate evidence of outperformance over existing Jump Models.

One avenue of future research is adapting the asymptotic properties of the Regularised K-Means model proven in (11) to the Regularised Jump Models. Some of these properties include consistency and strong consistency in terms of the Hausdorff distance (see (11) Theorem 6 and the proof in Appendix A. A.3).

We've proposed a new hyperparameter tuning method for Jump Models, based on the idea of clustering stability. There is scope for future research on developing and testing alternative hyperparameter tuning methods for Jump Models.

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

# A  SUPPLEMENTARY MATERIAL

# B  PARAMETER UPDATE EQUATIONS FOR REGULARISED JUMP MODEL CALIBRATION

**Proposition B.1.** *Suppose that we have an assignment of the elements of $Y$ into $K$ clusters $C_1, C_2, \ldots, C_K$. Let $|C_k|$ denote the number of elements in cluster $k$. Furthermore, let $\boldsymbol{M}$ be a $T \times K$ cluster assignment matrix with elements $m_{t,k} = \mathbf{1}_{\{\boldsymbol{y}_t \in C_k\}}$.*

*Let $\underline{\boldsymbol{\mu}}^*$ be the corresponding $K \times p$ matrix of cluster centres and $\underline{\mu}^*_{k,j}$ be the $(k, j)$ element of $\underline{\boldsymbol{\mu}}^*$ ($k^{th}$ cluster centre along the $j^{th}$ dimension). Keeping the cluster assignment matrix $\boldsymbol{M}$ fixed, solving (2.3) gives the following expressions for each penalty function $\mathcal{P}\left(\underline{\boldsymbol{\mu}}\right)$:*

$$\mathcal{P}_0\left(\underline{\boldsymbol{\mu}}\right): \ \underline{\mu}_{k,j} = \begin{cases} \underline{\mu}^*_{k,j} & \text{if } \|\boldsymbol{y}_{\cdot,j}\|^2 > \|\boldsymbol{y}_{\cdot,j} - \boldsymbol{M}\underline{\boldsymbol{\mu}}^*_{\cdot,j}\| + T\gamma \\ 0 & \text{otherwise} \end{cases} \tag{B.1a}$$

$$\mathcal{P}_1\left(\underline{\boldsymbol{\mu}}\right): \underline{\mu}_{k,j} = \max\left(0, 1 - \frac{T\gamma}{2|C_k||\underline{\mu}^*_{k,j}|}\right)\underline{\mu}^*_{k,j} \tag{B.1b}$$

$$\mathcal{P}_2\left(\underline{\boldsymbol{\mu}}\right): \underline{\mu}_{k,j} = \frac{1}{1 + \frac{T\gamma}{|C_k|}}\underline{\mu}^*_{k,j} \tag{B.1c}$$

$$\mathcal{P}_3\left(\underline{\boldsymbol{\mu}}\right): \underline{\mu}_{k,j} = \frac{1}{1 + \frac{T\gamma}{2|C_k||\underline{\boldsymbol{\mu}}_{\cdot,j}\|}}\underline{\mu}^*_{k,j} \quad \text{if } \underline{\boldsymbol{\mu}}_{\cdot,j} \neq \boldsymbol{0}, \tag{B.1d}$$

*where $\boldsymbol{y}_{\cdot,j}$ is the $j^{th}$ column of $Y$.*

*Proof.* Proof can be found in A.2. of the Appendix in (11). $\qquad\square$

**Remark B.2.** The update equations in (B.1) lend insight into the effects of each penalty function.

$\mathcal{P}_0$ leads to *hard-thresholding*, the process of setting to zero the elements of an input vector whose absolute values are lower than an input threshold value; elements whose absolute values exceed

the threshold are left unchanged. The threshold value in (B.1a) is the sum of $T\gamma$ and the WCSS contribution from each feature. The interpretation of (B.1a) is that variables are included in the clustering if it sufficiently contributes to a decrease of the WCSS. $\mathcal{P}_1$ leads to soft-thresholding, whereby some elements are set to zero and the rest are shrunk towards zero. This resembles the solution of $\mathcal{L}_1$-regularised (or Lasso) regression with orthonormal covariates. $\mathcal{P}_2$ is a ridge-type penalty that scales down each element of the array uniformly. $\mathcal{P}_2$ is the only penalty function that does not directly induce sparsity.

$\mathcal{P}_3$ does not have an updating equation since the right-hand side of (B.1d) includes the $\mathcal{L}_2$ norm of $\|\boldsymbol{\mu}_{.,j}\|$. The solution is thus found through an iterative algorithm.

| | $P$ | 15 | 30 | 60 | 150 | 300 |
|---|---|---|---|---|---|---|
| $\mu$ | **Model** | | | | | |
| 0.25 | Standard | 0.351 (0.06) | 0.338 (0.05) | 0.332 (0.05) | 0.343 (0.04) | 0.334 (0.03) |
| | Sparse | 0.349 (0.06) | 0.343 (0.05) | 0.343 (0.05) | 0.337 (0.04) | 0.336 (0.03) |
| | Regularised $\mathcal{P}_0$ | 0.354 (0.06) | 0.345 (0.05) | 0.336 (0.05) | 0.341 (0.05) | 0.334 (0.05) |
| | Regularised $\mathcal{P}_1$ | 0.359 (0.08) | 0.345 (0.06) | 0.341 (0.06) | 0.335 (0.04) | 0.344 (0.04) |
| | Regularised $\mathcal{P}_2$ | 0.343 (0.07) | 0.342 (0.06) | 0.340 (0.05) | 0.335 (0.05) | 0.344 (0.04) |
| | Regularised $\mathcal{P}_3$ | 0.348 (0.07) | 0.343 (0.06) | 0.339 (0.05) | 0.342 (0.05) | 0.334 (0.04) |
| 0.50 | Standard | 0.362 (0.10) | 0.368 (0.08) | 0.341 (0.07) | 0.341 (0.05) | 0.336 (0.04) |
| | Sparse | 0.376 (0.08) | 0.356 (0.08) | 0.345 (0.06) | 0.345 (0.05) | 0.344 (0.04) |
| | Regularised $\mathcal{P}_0$ | 0.358 (0.09) | 0.374 (0.09) | 0.365 (0.09) | 0.349 (0.07) | 0.339 (0.06) |
| | Regularised $\mathcal{P}_1$ | 0.362 (0.11) | 0.393 (0.13) | 0.387 (0.12) | 0.350 (0.09) | 0.349 (0.05) |
| | Regularised $\mathcal{P}_2$ | 0.380 (0.14) | 0.382 (0.14) | 0.389 (0.13) | 0.344 (0.07) | 0.360 (0.07) |
| | Regularised $\mathcal{P}_3$ | 0.383 (0.14) | 0.382 (0.14) | 0.374 (0.13) | 0.358 (0.08) | 0.354 (0.06) |
| 0.75 | Standard | 0.385 (0.15) | 0.401 (0.13) | 0.366 (0.10) | 0.353 (0.06) | 0.355 (0.06) |
| | Sparse | 0.380 (0.12) | 0.363 (0.11) | 0.364 (0.09) | 0.347 (0.07) | 0.335 (0.06) |
| | Regularised $\mathcal{P}_0$ | 0.418 (0.14) | 0.386 (0.14) | 0.389 (0.14) | 0.387 (0.13) | 0.408 (0.14) |
| | Regularised $\mathcal{P}_1$ | 0.418 (0.18) | 0.406 (0.18) | 0.432 (0.18) | 0.394 (0.17) | 0.458 (0.18) |
| | Regularised $\mathcal{P}_2$ | 0.400 (0.18) | 0.420 (0.16) | 0.447 (0.18) | 0.399 (0.18) | 0.447 (0.18) |
| | Regularised $\mathcal{P}_3$ | 0.407 (0.18) | 0.435 (0.19) | 0.391 (0.17) | 0.403 (0.15) | 0.468 (0.17) |
| 1.00 | Standard | 0.448 (0.20) | 0.423 (0.15) | 0.391 (0.13) | 0.357 (0.09) | 0.347 (0.08) |
| | Sparse | 0.408 (0.15) | 0.393 (0.14) | 0.363 (0.12) | 0.374 (0.11) | 0.347 (0.08) |
| | Regularised $\mathcal{P}_0$ | 0.423 (0.19) | 0.409 (0.18) | 0.464 (0.19) | 0.411 (0.16) | 0.494 (0.20) |
| | Regularised $\mathcal{P}_1$ | 0.483 (0.22) | 0.446 (0.24) | 0.524 (0.25) | 0.499 (0.24) | **0.564** (0.22) |
| | Regularised $\mathcal{P}_2$ | 0.495 (0.23) | 0.465 (0.24) | 0.523 (0.23) | 0.462 (0.23) | 0.498 (0.23) |
| | Regularised $\mathcal{P}_3$ | 0.444 (0.22) | 0.447 (0.24) | 0.507 (0.24) | 0.492 (0.22) | **0.597** (0.23) |

Table 2: Average BAC of state sequence (3 d.p.) for each Jump Model, across different values of $\mu$ and number of features $P$. Values in brackets are standard deviations (2 d.p.).

| | $P$ | 30 | 60 | 150 | 300 |
|---|---|---|---|---|---|
| $\mu$ | **Model** | | | | |
| 0.25 | Sparse | 0.499 (0.08) | 0.502 (0.05) | 0.502 (0.03) | 0.506 (0.03) |
| | Regularised $\mathcal{P}_0$ | 0.507 (0.04) | 0.498 (0.02) | 0.502 (0.01) | 0.501 (0.02) |
| | Regularised $\mathcal{P}_1$ | 0.511 (0.05) | 0.505 (0.03) | 0.496 (0.02) | 0.504 (0.03) |
| | Regularised $\mathcal{P}_3$ | 0.503 (0.05) | 0.506 (0.03) | 0.499 (0.03) | 0.498 (0.04) |
| 0.5 | Sparse | 0.509 (0.07) | 0.507 (0.05) | 0.509 (0.04) | 0.505 (0.04) |
| | Regularised $\mathcal{P}_0$ | 0.530 (0.05) | 0.529 (0.04) | 0.513 (0.03) | 0.511 (0.03) |
| | Regularised $\mathcal{P}_1$ | 0.560 (0.10) | 0.573 (0.10) | 0.535 (0.08) | 0.509 (0.04) |
| | Regularised $\mathcal{P}_3$ | 0.582 (0.10) | 0.588 (0.11) | 0.528 (0.06) | 0.513 (0.05) |
| 0.75 | Sparse | 0.538 (0.07) | 0.539 (0.08) | 0.531 (0.07) | 0.522 (0.06) |
| | Regularised $\mathcal{P}_0$ | 0.540 (0.07) | 0.538 (0.05) | 0.544 (0.07) | 0.568 (0.05) |
| | Regularised $\mathcal{P}_1$ | 0.594 (0.13) | 0.608 (0.14) | 0.621 (0.15) | **0.650** (0.13) |
| | Regularised $\mathcal{P}_3$ | 0.621 (0.15) | 0.603 (0.13) | 0.631 (0.16) | **0.665** (0.14) |
| 1.00 | Sparse | 0.574 (0.11) | 0.602 (0.12) | 0.587 (0.12) | 0.563 (0.10) |
| | Regularised $\mathcal{P}_0$ | 0.557 (0.10) | 0.549 (0.09) | 0.548 (0.08) | 0.624 (0.10) |
| | Regularised $\mathcal{P}_1$ | 0.622 (0.16) | 0.660 (0.18) | **0.695** (0.18) | **0.743** (0.17) |
| | Regularised $\mathcal{P}_3$ | 0.603 (0.14) | 0.629 (0.15) | 0.637 (0.16) | **0.782** (0.17) |

Table 3: Average BAC of relevant features (3 d.p.) for each Jump Model, across different values of $\mu$ and number of features $P$. Values in brackets are standard deviations (2 d.p.).

