# OpenReview forum: "Regularised Jump Models for Regime Identification and Feature Selection"
_ICLR.cc/2025/Conference — Submitted to ICLR 2025_

### Official Review · Reviewer_oMqY · 2024-10-28

**Soundness:** 1
**Presentation:** 1
**Contribution:** 1
**Rating:** 1
**Confidence:** 5

**Summary:**

This paper introduces a new set of models - the Regularised Jump Models, developed to enhance both regime identification and feature selection within the Jump Model framework for financial time series analysis.

**Strengths:**

The Regularised Jump Models adapt the Regularised K-means approach to the Jump Model framework and are designed to improve interpretability and accuracy over the Standard and Sparse Jump Models.

**Weaknesses:**

1. Although I am quite familiar with regime-switching, identification, and financial markets, the logical structure of this paper is challenging to follow. The introduction directly presents model definitions and the authors’ proposed improvements without first providing an overview of related work, the motivation, or the current state of research. This structure makes it difficult for readers to understand the unique contributions of this work and distinguish them from existing models. A clearer, more organized flow would greatly enhance readability and clarify the authors’ contributions.

2. The paper lacks a detailed motivation explaining why Regularised K-means, specifically, is optimal for adapting into the Jump Model framework. A comparison of Regularised K-means with alternative approaches could clarify why it is most suitable in this context.

3.  The paper briefly mentions Hidden Markov Models (HMM) and other regime-switching models but does not provide a thorough comparison, such as the latest model - RHINE: A Regime-Switching Model with Nonlinear Representation for Discovering and Forecasting Regimes in Financial Markets (SIAM SDM2024). Besides, although feature selection is central to the model, there is limited discussion on alternative feature selection methods in time series analysis.

4. The formulas (such as those in Eq. 1.6) lack detailed explanations for how each component, including the penalty terms $ P(\mu) $, specifically enhances feature selection and regime accuracy.

5. The experimental part does not demonstrate how regime identification or interpretability is achieved. Additionally, there are no actual experimental results presented in the main text, yet the pseudo-code of the algorithms takes up several pages.

6. Appendix A is left blank, and the purpose of Proposition B.1 in Appendix B is unclear—is it merely meant to illustrate the classic partitioning principle of K-means? This is a well-known concept in machine learning, and furthermore, the authors’ so-called “proof” is missing.

**Questions:**

As provided in weakness.

---

### Official Review · Reviewer_bmB3 · 2024-11-02

**Soundness:** 3
**Presentation:** 2
**Contribution:** 2
**Rating:** 3
**Confidence:** 4

**Summary:**

The paper introduces regularisation to the standard jump model, akin to the same modification in regularised k-means. The model is calibrated in the usual way. Hyperparameter tuning is via preferring clustering stability. Simulation results compare the regularised jump model to the standardised and sparse jump model.

**Strengths:**

The contribution of the paper is clearly presented.

**Weaknesses:**

Presentational Clarity:
There is a general lack of presentational clarity throughout. The reader has to guess what is intended much of the time (and if you're relying on this, then don't be unhappy if they can't follow):
* p.1. how does \bold{Y} relate to \bold{y}_t?
* p.2.\bar{\bold{\mu}} seems to be undefined.
* p.2. what mathematical object a cluster C_k entails is not clear.
* p.3 is \underline{\bold{\mu}} related to the \bold{\mu}_k? (c.f. line 2).
* p.3. Is \bold{\mu}_{\cdot,j} the same as or different to \bold{\mu}_j? It looks like there is different notation for the same object. (Same comment for \bold{y}_t and \bold{y}_{\cdot,j} in Appendix B.)
* p.6: It's not clear from what is written exactly what Step 3 in Algorithm actually means. Maybe it means (just guessing...) that one locates the largest (by some norm?) cluster centre magnitudes, take those observations allocated to those clusters, and then perform K-means clustering on these observations only? This detail (or whatever the detail actually is) really needs to be clearer.
* p.7 para 3: The  meaning of this notation is not clear. \psi takes a single p-dimensional vector X and returns a cluster ID integer. What is \Psi in relation to this? And why does \Psi have some parameter \boldysymbol{\theta} as a condition and \psi does not?
* Following from the previous point: this becomes more confusing when the clustering distance function d can seemingly take both \psi and \Psi as inputs (definitions 3.1 and 3.2), when these are clearly different objects.
* d in Definition 3.1 seems to be dependent on X and Y, which are random variables. This makes d a random variable. Is this correct and the paper's intention? Distances are typically deterministic.
* This is related to the expectation in (3.1). What is this expectation with respect to? Z_1 and Z_2 are draws from F. Once drawn and put into (3.1) they are fixed. So what is the expectation with respect to?
* Proposition B1: is Y here related to \bold{Y} in the main text? Is M related to \Psi in the main text?


Contribution:
a) The overall contribution seems fairly incremental. The paper adds a standard regularisation penalisation term to the jump model objective function in exactly the same way that a standard regularisation penalisation term is added to k-means. To be clear: there's nothing wrong with this, but it's a fairly low contribution (and not ICLR standard), and the simulations presented were not particularly convincing in support of this contribution (see more below).
b) There is no real motivation presented at the start of the paper for why the proposed work is needed or has value. Typically one would expect to see a strong justification to solve the problem, and a literature review discussing what has currently been done, and their strengths and weaknessess. This then allows the paper to present the justification for the work it contains in a clear and as strong as possible light, and against the current state of the art. This is not present in the current paper, which just lists things that were done.
c) Conclusion paragraph 2: Given the paper's incremental contribution, it would have been helpful to include the mentioned theoretical results "future work" in the current paper, though these results would likely have been standard extensions of existing results.


General Comments:
a) Section 2.2: S is introduced as a soft thresholding operator. S is also used as the vector of states (notation clash).
b) Section 2.2: Presumably Algorithm 2 should be referenced in here somewhere?
c) Regardless, algorithm 2 (sparse jump model calibration) is not really used in the development of the proposed regularised jump model (nor its implementation), so its not clear why it should be included on p.6 taking up lots of space that could be used  for improving the current paper. Suggest it is moved to supporting information at best.
d) Section 2.3: The strategy in publication [11] of repeating the optimisation with 7 initial state sequences seems alarmingly ad hoc and situation specific/dependent, and one might be wary of following such a strategy in general, as well as recommending it as a basis for new work. It should also be made clearer how this initialisation strategy "incorporates potential sparsity in the initial cluster centres", and indeed why there is any hope that this is likely to get the user anywhere close to the global optimum of the target function. (Note that "most optimal" in this paragraph should just be "optimal".)
e) P.6. Algorithm 3: does Step 3 include (with the "100% largest d_j") an exact repeat of Step 1?
f) Clustering stability (definition 3.2) seems to prioritise the tendency of producing similar clusters over (say) parsimony. Is it clear that this is the best approach? (E.g. an algorithm that gives clusters of (1,1,1, ..., 1) deterministically each time without variation will be preferred over any other.) It would be good to understand a little more (via a discussion) why this prioritisation is aligned with the goals of the analysis.


Simulation Study:
It was difficult to take much from this section, as written. It would benefit from a complete re-write (and expansion).
a) The simulation study details are all in another paper [9]. The reader is not going to go and read that paper so that they can understand what has been done in this paper. All relevant detail should be contained in the paper it is used in. As such, it was difficult for this reader to take much from this simulation study.
b) This includes algorithm/simulation settings etc. Was K fixed at the true value of K each time? How would it perform if it was not? etc. We have no idea.
c) Section 4.1 para 1: BAC is not defined (or at least, this reader couldn't find it), and so can't be understood. \mu here is not bold face. Previously \mu was \underline{\bold{\mu}} was the matrix of the cluster centres \bold{\mu}_k. Are these related? \mu seems to be a single integer in Tables 2 and 3. This really all makes no sense.
d) Section 4.1 para 4: \mu seems to impact algorithm performance, which seems incredible if it is a cluster centre.
e) The captions in the tables are not sufficient for the reader to know what the content of the tables is.
f) p.9 para 2: Its not clear what calculation was performed here.
g) Much of the discussion in Section 4.1 is of the form of empirical observations on tables of numbers, without attempting real understanding of what this means as general principles of algorithm performance. That is, we only have very specific, empirical observations.

**Questions:**

See Weaknesses.

---

### Official Review · Reviewer_Y32j · 2024-11-02

**Soundness:** 2
**Presentation:** 1
**Contribution:** 1
**Rating:** 1
**Confidence:** 3

**Summary:**

The paper proposed a Regularised Jump Model, an extension of the existing models based on the Jump Model Framework.This class of models aims to accurately identify different regimes of financial markets in particular through improved feature selection.  The proposed method relies on regularized K-means algorithm to achieve regularization. Several different penalty terms are considered for regularization (L0, Lasso, Ridge, Group-Lasso) and three different Jump Models (Stadard, Sparse, Regularized – the latter proposed in this paper) are compared in a simulation study.

**Strengths:**

The authors identify a relevant problem in Jump Models for financial time series. With more development, this work could offer useful insights into this type of models and new ways to improve model regularization.

**Weaknesses:**

- There is very limited information on related work;
- The structure of the paper is not always clear;
- The adaptations to the existing model are somewhat minor, and intuition isn't always provided on how the proposed methodology;
- There is no real-world data application/comparison;
- The proposed model does not always outperform existing models (standard jump model and sparse jump model) and no recommendation is given on which one should be chosen in which situations (same holds for the different penalties in the proposed approach);

**Questions:**

1) Introduction should include related work and motivation of why these models are relevant. At the moment, the introduction immediately introduces the model. It is impossible for a general reader of the conference to place this in a context.
2) How are the jump models related to Hidden Markov Models? What is the advantage of using them?
3) The paper suggests that more interpretable feature selection could be made with the proposed approach. However, no results are given for this. The only measure used for performance evaluation is BAC.
4) Providing a comparison on computational time would be relevant.
5) There is no example with real-world data or any illustration of how this is relevant in practice.
6) For the simulation study, the paper refers to another one without providing any details or intuition on the set-up. It would be nice if the motivation (including the motivation for the choices of the simulation study) was more self-contained.
7) It's unclear when to use the sparse jump model and when to use the proposed regularized jump model, and the differences in performance are marginal, and sometimes sparse models outperform the proposed regularized jump model. Clear recommendations should be given on when to use which one and what leads to differences in performance.

---

### Official Review · Reviewer_8qTi · 2024-11-04

**Soundness:** 2
**Presentation:** 1
**Contribution:** 2
**Rating:** 3
**Confidence:** 3

**Summary:**

Proposing algorithms for jump models while also utilizing feature selection.

**Strengths:**

- Inclusion of multiple penalty terms in the proposed method.

-

**Weaknesses:**

- Many typos throughout the manuscript.

- Math notations not defined, such as $\bar{\mu}$ in (1.2).

- Many details of the algorithms are missing.

- Real data application is missing with lowers the practical importance of the proposed methodologies.

- It seems that the total number of states K is assumed to be known. This is a restrictive assumption, especially in real data applications.

**Questions:**

- What is the state sequence role in eq. (1.2)?

- The maximization is over what in eq. (1.2)?

- Is the total number of jumps/ clusters K assumed to be known?

- How to choose the number of states K in practice?

---

### Meta-Review · Area_Chair_JFKe · 2024-12-18

**Metareview:**

The paper proposes an extension of the jump model framework, aimed at improving regime identification and feature selection in financial time series. These models incorporate regularization techniques (e.g., Lasso, Ridge, Group-Lasso) to enhance interpretability and performance over the Standard and Sparse Jump Models. Through simulations, the authors claim that the proposed models outperform existing ones in both regime identification and feature selection. However, reviewers identified critical issues. The paper lacks clarity, with poorly defined mathematical notations and an unclear structure. Motivation, comparison to related work, and practical relevance are insufficiently addressed. The contribution is seen as incremental, adding regularization techniques without sufficient justification or novelty. Real-world applications are absent, and simulation details are incomplete, limiting the paper's practical impact and scientific rigor. The reviewers generally agree that the work requires significant revision, including better organization, clearer explanations, comprehensive comparisons, and practical validations. Overall, I believe that the submission is poor, needing further work until these issues are addressed.

**Additional Comments On Reviewer Discussion:**

The authors did not respond to the reviewers.

---

### Decision · Program_Chairs · 2025-01-22

Reject